# The analgesic efficacy of pericapsular nerve group block in patients with intertrochanteric femur fracture: A randomized controlled trial

Mingjian Kong[1]*, Yan Tang[2], Fei Tong[1‡], Hui Guo[1‡], Xin Lei Zhang[1], Lei Zhou[3], Hua Ni[4], Bin Wang[3], Yunqing Liu[1], Jindong Liu[2]

1 Department of Anesthesiology, The Second Affiliated Hospital of Xuzhou Medical University, Xuzhou, Jiangsu Province, China, 2 Department of Anesthesiology, The Affiliated Hospital of Xuzhou Medical University, Xuzhou, Jiangsu Province, China, 3 Department of Orthopaedics, The Second Affiliated Hospital of Xuzhou Medical University, Xuzhou, Jiangsu Province, China, 4 Department of Rehabilitation, The Second Affiliated Hospital of Xuzhou Medical University, Xuzhou, Jiangsu Province, China

☯ These authors contributed equally to this work.
‡ FT and HG also contributed equally to this work.
* mjkong@126.com

**Data Availability Statement:** The data that support the findings of this study are openly available in the following repositories:

## Abstract

### Background

The aim of this study is to evaluate analgesic efficacy of pericapsular nerve group (PENG) block in patients with intertrochanteric femur fracture (IFF).

### Methods

This double-blinded randomized controlled trial in patients with IFF scheduled for proximal femoral nail antirotation (PFNA) between December 2020 and November 2021. The primary outcome was VAS scores during exercising at 6 h after surgery; secondary outcomes were pain during exercising and rest, intraoperative dose of remifentanil, cumulative dose of post-operative fentanyl, postoperative analgesia satisfaction scores, and ratio of quadriceps weakness.

### Results

A total of 50 patients were randomly divided into PENG block group (n = 25) or fascia iliaca compartment block (FICB) group (n = 25). Exercising VAS scores at 6 h after surgery were significantly lower in PENG block group than that in FICB group (2 (2, 4) vs. 6 (4, 7), $P <$ 0.001). The intraoperative dose of remifentanil and cumulative dose of postoperative fentanyl by patient-controlled intravenous analgesia within 24 h after surgery in PENG block group were significantly lower than in FICB group (both $P <$ 0.001). Postoperative analgesia satisfaction scores in PENG block group were significantly higher than those in FICB group ($P =$ 0.016). The ratio of quadriceps weakness at 6 h after surgery was significantly higher in FICB group than PENG block group (48% vs. 0%, P < 0.001).

https://figshare.com/account/home#/data. The DOI
is: https://10.6084/m9.figshare.21214463. The
data are also in the Supporting information files.

**Funding:** The author sreceived no specific funding
for this work.

**Competing interests:** The authors have declared
that no competing interests exist.

## Conclusions

Compared to FICB, ultrasound-guided PENG block may provide better postoperative pain relief in patients with IFF, with less pronounced quadriceps weakness.

## Introduction

Intertrochanteric femur fracture (IFF) is a common traumatic event in elderly patients, where it accounts for approximately 45% to 50% of all hip fractures [1]. IFF often causes severe pain, up to 9 points in the movement-associated median pain score [2], limiting the activities of patients even after surgery, and increasing the incidence of deep venous thrombosis of the lower extremities, pulmonary infection and mortality [3]. Therefore, effective analgesia could improve postoperative quality of life, promote early postoperative functional exercises and reduce complications.

Peripheral nerve blocks (PNB) were shown to have less influence on the hemodynamics, respiration function and consciousness, compared to systemic pain relief options. Notably, PNBs are recommended as the first-line analgesia program for hip surgeries, with the ability to reduce doses of opioids, shorten the postoperative recovery time, and decrease the risk of pneumonia [4]. Among the most common PNB locations, fascial iliaca compartment is a space between the iliopsoas muscle and the iliac fascia, where three principal lumbar plexus nerves of the thigh are located, namely the femoral, lateral cutaneous, and obturator nerve. Fascia iliaca compartment block (FICB) is safe and simple, commonly used for perioperative analgesia of hip fracture, which in theory can simultaneously block femoral nerve, lateral femoral cutaneous nerve and obturator nerve, to obtain distal lumbar plexus block effect. Compared with intravenous analgesia, FICB can alleviate postoperative pain and decrease morphine consumption after hip surgery [5, 6]. However, it has been reported that in some cases FICB does not provide the adequate analgesia or decrease in the opioid consumption after total hip arthroplasty [7], due to the failure of FICB to block obturator nerve [8].

Pericapsular nerve group (PENG) block, which targets the articular branches of femoral nerve, obturator nerve and accessory obturator nerve, was firstly introduced by Girón-Arango in 2018 [9]. PENG block have a potential for hip fracture, with the ability to reduce the median of dynamic pain score by 7 points in hip fracture patients which is superior to other PNBs [9]. Recently, several case reports have shown that PENG block provides a sufficient analgesia during hip fracture surgeries without a potential influence on the quadriceps muscle strength [10, 11], but prospective studies on the analgesic efficacy of PENG block in comparison to other, more traditional PNB approaches are still needed.

Therefore, the primary objective of this study was to compare the analgesic effects of ultrasound-guided PENG block and FICB in IFF patients undergoing proximal femoral antirotation (PFNA) under general anesthesia. A secondary objective was to compare the effects of PENG block and FICB on postoperative muscle strength in patients. At the same time, the related complications of PENG block were observed, and the safety of its clinical application was discussed.

## Methods

### Study design and subjects

This manuscript adheres to the applicable CONSORT guidelines. This RCT included patients with IFF diagnosed by radiography criteria proposed by Marsh [12], and receiving PFNA

under general anesthesia at the department of Orthopaedics in the Second Affiliated Hospital of Xuzhou Medical University between December 2020 and November 2021. The study was approved by the Ethics Committee of The Second Affiliated Hospital of Xuzhou Medical University ([2020] 102901) and written informed consent was obtained from all subjects participating in the trial. Before patient enrolled, the study was registered at chictr.org.cn (identifier: ChiCTR2000039749, principal investigator: Fei Tong, Date of registration: November 7, 2020). The ethics committee supervises the whole process of the experiment and ensures the safety of patients.

Inclusion criteria: 1) 65–85 years of age; 2) American Society of Anesthesiologists (ASA) grade I-III; 3) preoperative resting visual analogue scale (VAS) scores $\geq$ 4 points [13].

Exclusion criteria: 1) Unable to orally communicate; 2) Body mass index (BMI) $> 30 \text{ kg/m}^2$ or $<18.5 \text{kg/m}^2$; 3) Allergic to the study drugs; 4) Liver or renal insufficiency; 5) Opioid addiction or dependence; 6) Cognitive impairment; 7) Neuromuscular disorders; 8) Local infection of nerve block; 9) Surgery time $> 2$ hours (Fig 1).

## Randomization and blinding

Patients were randomly assigned to the PENG group and the FICB group at 50% of each group according to a computer-generated random number table. This study was blinded for the anesthesiologists, data collectors, statistical analysts, and patients. A staff member A, who was not involved in the following research, delivered the card with the grouping in the opaque envelope with the random number on the cover. After the patient enters the operating room, A handed over the envelope to operator B. Upon opening the envelope, B clarified the method of nerve block and performed corresponding operations. B was only responsible for the nerve block operation, and has no knowledge regarding the content of the following research. Before and after the completion of the B operation, the special data collector C will collect the pain score and other data. C was blind to the classification of the nerve block, the subsequent anesthesia work was performed by an uninformed anesthesiologist. All data of postoperative patients were collected by C, and the entire procedure was blinded to patients.

## Intervention

Patients in both groups were fasting for 8 h, with drinking prohibited 2 h prior to the surgery. Electrocardiography, pulse oxygen saturation, invasive radial arterial pressure, and bispectral index (BIS) were continuously monitored during the surgery.

Patients in PENG group received PENG block as described by Girón-Arango [9]. They were placed in the supine position, and a low-frequency curvilinear ultrasound probe (2–6 MHz, Sonosite) was horizontally placed above the anterior superior iliac spine and moved toward the pubic bone. After the anterior inferior iliac spine was visualized, the probe was rotated parallel to the pubic branch until the anterior inferior iliac spine, iliopsoas eminence, femoral artery, iliopsoas muscle and its tendons were clearly visualized. A 22G puncture needle was punctured outside-in under the guidance of ultrasound. 2 mL of normal saline was injected until the tip of the puncture needle reach the musculofascial plane between the psoas tendon anteriorly and the pubic ramus posteriorly to identify the correct location of the tip, followed by injection of 30 mL of 0.375% ropivacaine (Naropin, AstraZeneca, Fig 2).

Patients in FICB group received FICB as described by Dolan [14]. They were placed in the supine position, and a high-frequency line probe (6-13MHz, Sonosite) was placed near the inguinal ligament, which clearly visualized the femoral artery, femoral vein, iliac muscle and sartorius muscle. At 1 cm under the line connecting the pubic tubercle and anterior superior iliac spine, a 22G puncture needle was punctured outside-in. 2 mL of normal saline was

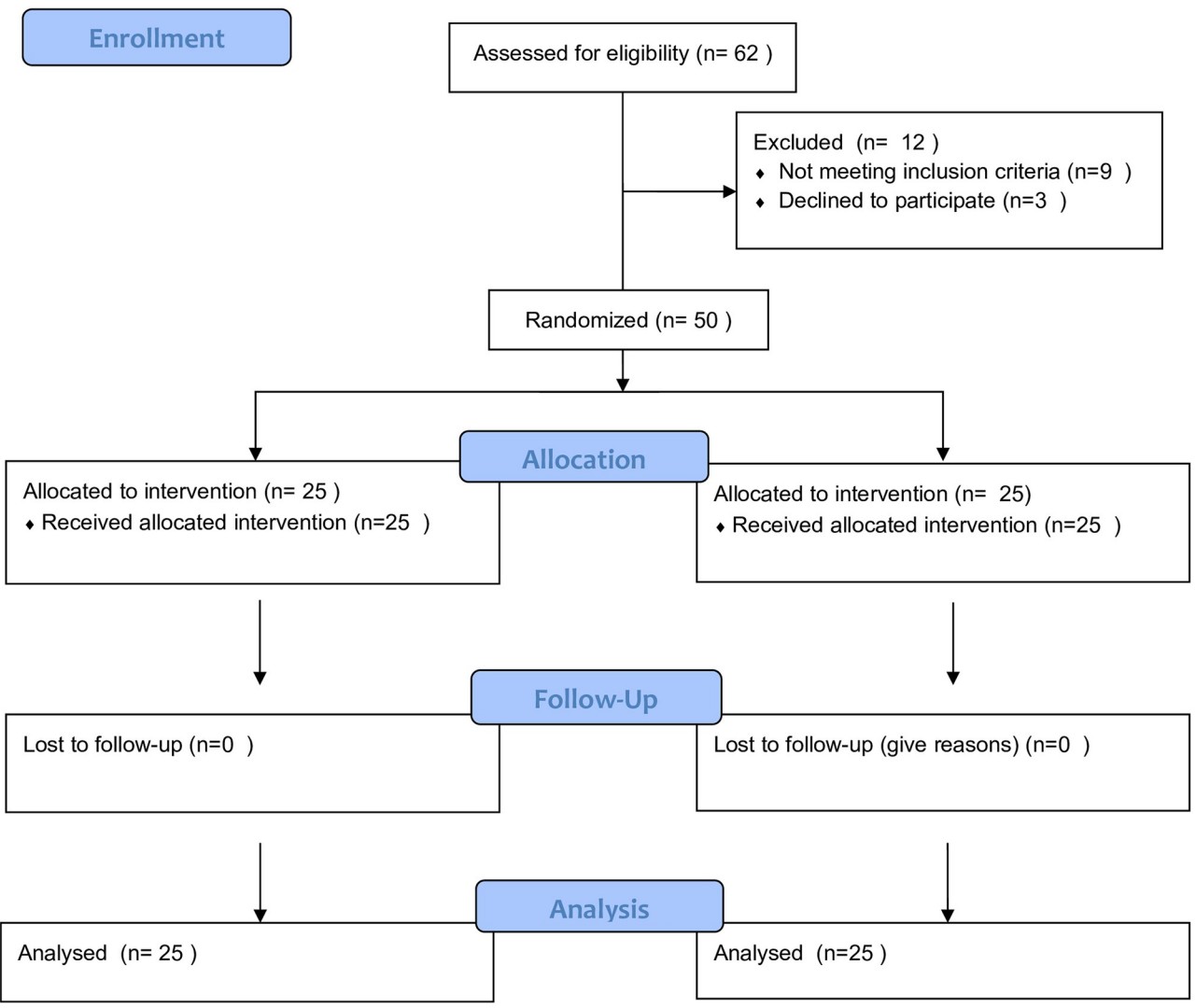

**Fig 1. Flow chart of the present study.**

injected until the tip of the puncture needle reach the musculofascial plane between the iliac fascia and iliopsoas muscle to identify the correct location of the tip, followed by injection of 30 mL of 0.375% ropivacaine (Fig 3).

Patients in both groups were intravenously administrated with 1 μg/kg fentanyl, 1.5–2 mg/kg propofol and 0.1 mg/kg atracurium for induction of anesthesia. General anesthesia was maintained by administration of 1.0–1.5% sevoflurane and 0–0.3 μg/kg/min remifentanil (according to the BIS and hemodynamics). All patients were administrated with 15 ml of 0.375% ropivacaine for local infiltration anesthesia before suturing. After removing the laryngeal mask in the operating room, they were sent to the post-anesthesia care unit (PACU).

In the surgical ward, as addition to patient-controlled intravenous analgesia (PCIA) pump with a 10-μg fentanyl bolus and a 20-min lockout period with no background, all patients received regular intravenous Flurbiprofen axetil 50 mg every 24 hours during 48 hours.

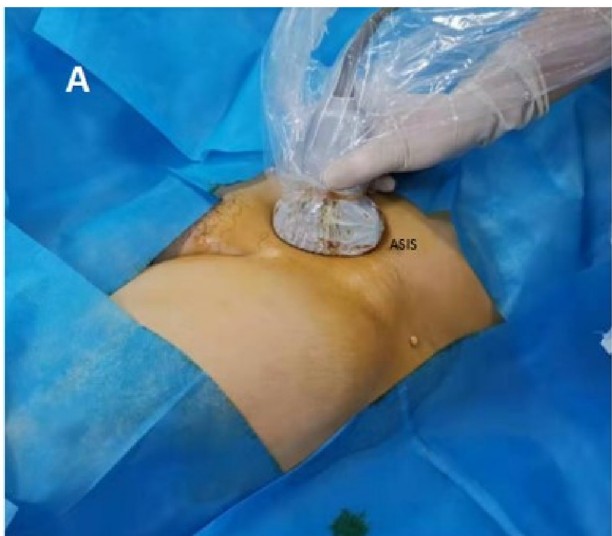

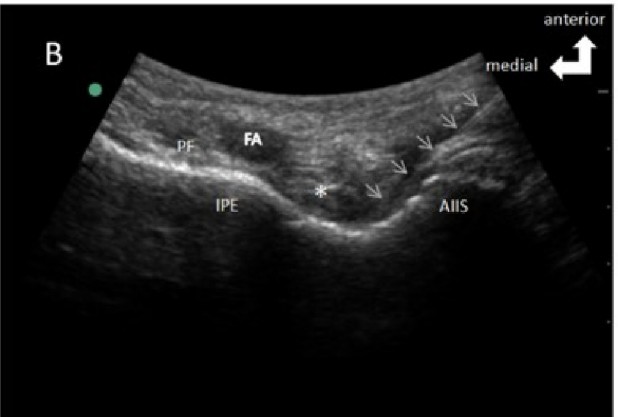
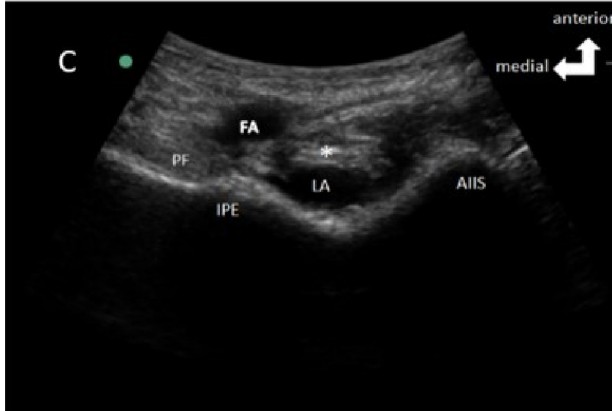

**Fig 2. Representative image and ultrasound scans of the patient receiving PENG block.** (A) Patients' position and the placement of the probe. (B) Ultrasound-guided tracing of the puncture needle (white arrow). (C) Deposition of local anesthetics. * Iliopsoas tendon is marked on Fig 1B and 1C. PENG, pericapsular nerve group; AIIS, anterior inferior iliac spine; IPE, iliopubic eminence; FA, femoral artery; LA, local anesthetics.

## Outcomes

The primary outcome was the postoperative pain during exercising (passive elevation of the lower limb at 15°) at 6 h (T2) after surgery. Postoperative pain was evaluated by a VAS ranging from 0 to 10, in which 0 indicates no pain and 10 indicates extreme pain; a pain score of 1–3 was considered to indicate mild pain [15].

The secondary outcomes included VAS scores at rest and during exercise at different time points, dosage of remifentanil and fentanyl, rate of quadriceps motor block, postoperative analgesia satisfaction scores and block-related complications and postoperative adverse effects.

The VAS scores at rest and during exercise at different time points referred to scores before nerve block (T0), 30 min after nerve block (T1), 24 h (T3), 48 h (T4) and 72 h (T5) postoperatively and the rest VAS scores at 6 h postoperatively.

Intraoperative dose of remifentanil, postoperative cumulative dose of fentanyl at 24 h and 24~48 h were recorded, as well as block-related complications and postoperative adverse effects, such as postoperative nausea, vomiting, dizziness, delirium, urinary retention and deep

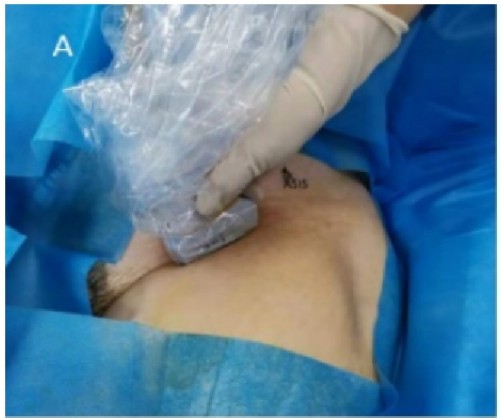

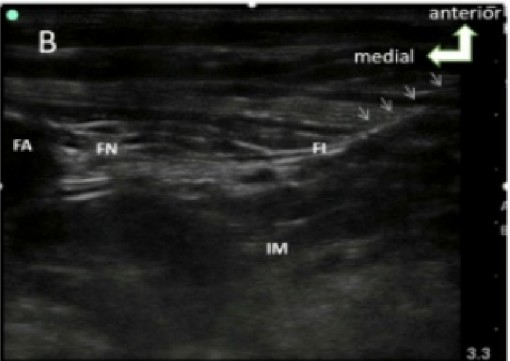
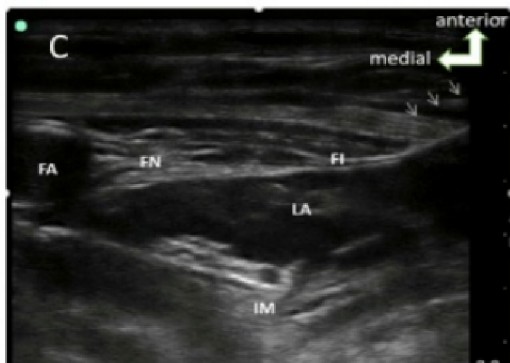

**Fig 3. Representative image and ultrasound scans of the patient receiving FICB.** (A) Patients' position and the placement of the probe. (B) Tracing of the puncture needle (white arrow). (C) Deposition of local anesthetics. FICB, fascia iliac compartment block; FA, femoral artery; FN, femoral nerve; FI, Iliac fascia; IM, iliacus muscle; LA, local anesthetics.

venous thrombosis of lower extremities. Postoperative analgesia satisfaction scores were graded as 0 point, dissatisfied; 1 point, average; 2 points, fair; 3 points, satisfied; and 4 points, very satisfied.

Quadriceps motor block was assessed using knee extension at 6 h after surgery. Knee extension was evaluated in a supine position with the patient's hip and knee flexed at 45˚ and 90˚, respectively. Knee extension was graded according to a 3-point scale: 0 = no block, 1 = paresis (decreased ability to extend the leg), 2 = paralysis (inability to extend the leg) [16].

## Statistical analysis

The preliminary pilot study involving 10 elderly patients with intertrochanteric femur fracture showed that the VAS scores at 6 h after surgery were $3.4 \pm 1.9$ (mean ± SD, PENG block group) and $5.4 \pm 2.1$ (mean ± SD, FICB group) respectively. Based on the pilot study, and considering a 20% of dropout rate ($\alpha = 0.05$, power of test = 90%), we determined that a sample size of 25 patients per group.

Normal distribution of continuous data was assessed using the Shapiro-Wilk test, the Levene test was used to verify homogeneity. Continuous data with normal distribution were expressed as mean ± standard deviation (SD) and compared using the Student's *t*-test; Otherwise, data were expressed as the median and quartiles, and compared using the Mann-Whitney U-test. Categorical data were expressed as numbers and percentages, and compared using the Chi-square test or Fischer's exact test. Statistical analyses were performed using IBM SPSS

Statistics 25 (IBM, Armonk, NY, USA). Two-tailed *P* value < 0.05 was considered statistically significant.

## Results

62 elderly patients were screened for eligibility for the study by the research assistant (Fig 1), 12 patients were excluded or declined participation. Finally, 50 patients were randomly divided into PENG block group (n = 25) and FICB group (n = 25), with the mean age of 73.4 ± 5.9 and 72.8 ± 4.8 years respectively (Table 1).

As demonstrated in Table 2, exercising VAS scores in PENG block group at 6 hour after surgery were significantly lower than those of FICB group (2(2,4) VS.6(4,7)), (*P* < 0.001). The resting VAS scores in PENG block group at T1-T3 were significantly lower than those of FICB group (all *P* < 0.001) as well as exercising VAS scores at T1-T4 (all *P* < 0.001, Table 2).

The intraoperative remifentanil and cumulative postoperative fentanyl doses by PCIA within 24 h after surgery in PENG block group were significantly lower than those in FICB group (*P* < 0.001). However, no significant difference in the cumulative dose of postoperative fentanyl at 24–48 h was detected between groups (*P* = 0.396, Table 3). Compared with that of FICB group, postoperative analgesia satisfaction scores were significantly higher in PENG block group (*P* = 0.016, Table 3).

The incidence of quadriceps motor block was significantly higher in FICB group than that of PENG block group (48% vs. 0%, *P* < 0.001). Complications related to the nerve blocking such as local anesthetic poisoning, nerve injury, hematoma and hip infection were not reported in the two groups. We did not detect significant differences in the incidences of post-operative nausea, vomiting, dizziness, delirium, urinary retention and deep venous thrombosis of lower extremities between groups (all *P* > 0.05, Table 4).

## Discussion

In our study, the primary analysis is analgesic effect of two different nerve block methods in elderly patients after PFNA. The secondary analysis is the effect of different nerve block

**Table 1. Characteristics of patients.**

| Characteristics | PENG block group (n = 25) | FICB group (n = 25) | P |
|---|---|---|---|
| Age (year, mean ± SD) | 73.4 ± 5.9 | 72.8 ± 4.8 | 0.627 |
| Male/female [n (%)] | | | 0.569 |
| Male | 12 (48) | 10 (40) | |
| Female | 13 (52) | 15 (60) | |
| Height (m, mean ± SD) | 1.66 ± 0.07 | 1.64 ± 0.06 | 0.417 |
| Body weight (Kg, mean ± SD) | 70.9 ± 9.0 | 69.8±8.4 | 0.663 |
| BMI (Kg/m2, mean ± SD) | 25.6 ± 1.9 | 25.7 ± 1.9 | 0.906 |
| ASA grade II/III [n (%)] | | | 0.508 |
| Grade II | 7 (28) | 5 (20) | |
| Grade III | 18 (72) | 20 (80) | |
| VAS scores at rest [M (IQR)] | 5 (5, 6) | 6 (5, 6.5) | 0.622 |
| VAS scores on exercise [M (IQR)] | 9 (7, 10) | 8 (7, 9) | 0.417 |
| Surgery time (min, mean ± SD) | 87.7 ± 20.7 | 87.6 ± 18.7 | 0.989 |
| T1-T2 time (min, mean ± SD) | 477.72±19.67 | 477.64±17.89 | 0.988 |

PENG, pericapsular nerve group block; FICB, fascia iliac compartment block; BMI, body mass index; ASA, American Society of Anesthesiologists; VAS, visual analogue scale; M, median value; IQR, interquartile range.

**Table 2. VAS scores in patients receiving PENG or FICB at different time points.**

| Characteristics | Time points | PENG block Group (n = 25) | FICB Group (n = 25) | P |
|---|---|---|---|---|
| VAS scores at rest [M (IQR)] | T0 (before nerve block) | 5 (5, 6) | 6 (5,6.5) | 0.622 |
| | T1 (30 min after block) | 2 (1,2.5) | 3 (2,4) | 0.001 |
| | T2 (6 h postoperatively) | 2 (1,2) | 3 (2,4) | <0.001 |
| | T3 (24 h postoperatively) | 2 (1,3) | 3 (3,4) | 0.001 |
| | T4 (48 h postoperatively) | 3 (2,4) | 4 (3,4) | 0.051 |
| | T5 (72 h postoperatively) | 3 (2,4) | 3 (2,4) | 0.442 |
| VAS scores on exercise [M (IQR)] | T0 (before nerve block) | 9 (7,10) | 8 (7,9) | 0.417 |
| | T1 (30 min after block) | 3 (2,3) | 5 (3,6.5) | <0.001 |
| | T2 (6 h postoperatively) | 2 (2,4) | 6 (4,7) | <0.001 |
| | T3 (24 h postoperatively) | 3 (2,4.5) | 5 (4,7) | <0.001 |
| | T4 (48 h postoperatively) | 3(2,5) | 5 (4,6) | <0.001 |
| | T5 (72 h postoperatively) | 5 (3,6) | 5 (3.5,6) | 0.670 |

VAS, visual analogue scale; M, median value; IQR, interquartile range; PENG, pericapsular nerve group block; FICB, fascia iliac compartment block

methods on quadriceps femoris muscle strength. We also analyzed the safety of anesthesia related complications after nerve block surgery.

Compared to scoring before blocking, PENG block reduced exercising VAS pain scores by 7 points at 6 h after surgery (and 6 points 30 min after blocking), while FICB reduced exercising VAS pain scores by 2 and 3 points respectively at the same time points. The results suggested that patients who received PENG block had demonstrated significantly lower resting and exercising VAS scores and less opioid consumption during and after surgery than those who received FICB.

PENG block has been reported to provide adequate alagesia for fracture and dislocation of the hip joint, as well as total hip arthroplasty [9–11, 17]. On the other hand, FICB performed via anterior approach to the lumbar plexus, was reported to provide only moderate pain relief for hip surgery [18]. The reason for that lies in both magnetic resonance and autopsy studies confirming that FICB could not cover the obturator nerve [8, 19], while femoral obturator nerve block provides better analgesic effects than FICB for hip fracture [13]. In contrast, PENG targets obturator nerve and accessory obturator nerve specifically, as was demonstrated in previous cadaveric study [20]. In present study the majority of patients who received FICB suffered moderate to severe postoperative pain during exercising, indicating that FICB analgesia might be inadequate for IFF. These results are in line with the previous case reports

**Table 3. Intraoperative and postoperative doses of opioid analgesics, and postoperative analgesia satisfaction scores.**

| Characteristics | PENG Group (n = 25) | FICB Group (n = 25) | P |
|---|---|---|---|
| Intraoperative dose of remifentanil [μg, M (IQR)] | 102 (95.5, 122.5) | 186 (148, 215.5) | <0.001 |
| Cumulative dose of postoperative fentanyl [μg, M (IQR)] | | | |
| 24 h postoperatively | 0 (0, 20) | 40 (20, 60) | <0.001 |
| 24–48 h postoperatively | 40 (40, 60) | 40 (20, 60) | 0.396 |
| Postoperative analgesia satisfaction scores [M (IQR)] | 3 (3, 4) | 3 (2, 3) | 0.016 |

PENG, pericapsular nerve group block; FICB, fascia iliac compartment block; M, median value; IQR, interquartile range.

**Table 4. Incidences of quadriceps motor block and adverse events.**

| Characteristics [n (%)] | PENG Group (n = 25) | FICB Group (n = 25) | P |
|---|---|---|---|
| Quadriceps motor block | | | <0.001 |
| No block | 25 (100) | 13 (52) | |
| Paresis | 0 (0) | 8 (32) | |
| Paralysis | 0 (0) | 4 (16) | |
| Nausea | 1 (4) | 4 (16) | 0.349 |
| Vomiting | 0 (0) | 0 (0) | 1 |
| Dizziness | 1 (4) | 2 (8) | 1 |
| Delirium | 0 (0) | 0 (0) | 1 |
| Urinary retention | 0 (0) | 0 (0) | 1 |
| Deep venous thrombosis of lower extremities | 0 (0) | 0 (0) | 1 |

PENG, pericapsular nerve group block; FICB, fascia iliac compartment block.

suggesting that the inadequate analgesic effect of FICB is explained by the failure of obturator nerve block.

The preservation of quadriceps muscle strength is beneficial for early restoration of the daily function, as well as for minimizing the risk of fall during postoperative exercises [21]. As revealed in the previous study, PENG block did not weaken the quadriceps muscle strength, while FICB caused the decrease in 61% of patients [22]. While Aliste reported that the incidence of quadriceps movement block caused by PENG block 6h after surgery was 25%, while that of suprainguinal fascia iliaca compartment block (SFICB) was 85%. However, higher local anesthetic concentration was used in PENG block in this study than in this study [23]. Another study by Yu et al. [24] reported 2/100 cases of accidental quadriceps weakness following PENG block, which is caused by the spread of local anesthetic to the femoral nerve and iliac fascia space due to the medial and superficial position of the tip of puncture needle. As revealed in the present study, PENG block did not weaken the quadriceps muscle strength, while FICB caused the decrease in 48% of patients. It might be explained by the fact that PENG block targets the articular branch of femoral nerve, rather than the whole nerve, and as a result, the quadriceps muscle strength can be preserved.

This study has several limitations. Firstly, the sample size of this study was relatively small, and not powered to observe some rare complications. Secondly, adductor strength, hip joint functional recovery and length of hospital stay data were not collected. And finally the dose of fentanyl was controlled by patients, which may cause potential bias.

In conclusion, compared to FICB, ultrasound-guided PENG block provides better postoperative pain relief in patients with intertrochanteric femur fracture, with less pronounced quadriceps weakness. PENG block may be a good alternative analgesic method for hip fractures, and further large scale randomized clinical trials are needed.

## Supporting information

**S1 Checklist. CONSORT 2010 checklist of information to include when reporting a randomised trial\*.**
(DOC)

**S1 Data.**
(XLSX)

## Acknowledgments

We are very grateful to the **S**cience & Technology Projects in Xuzhou for their funding and to all authors who have made contributions to the study work.

**Declaration**: All authors listed meet the authorship criteria according to the latest guidelines of the International Committee of Medical Journal Editors, and that all authors are in agreement with the manuscript.

## Author Contributions

**Conceptualization:** Lei Zhou.

**Data curation:** Mingjian Kong, Yan Tang, Hui Guo, Xin Lei Zhang, Hua Ni, Bin Wang.

**Formal analysis:** Mingjian Kong, Fei Tong, Hui Guo, Xin Lei Zhang.

**Investigation:** Xin Lei Zhang.

**Methodology:** Yan Tang, Fei Tong, Yunqing Liu.

**Project administration:** Bin Wang.

**Supervision:** Hui Guo, Lei Zhou.

**Visualization:** Mingjian Kong.

**Writing – original draft:** Mingjian Kong, Yan Tang, Xin Lei Zhang, Yunqing Liu, Jindong Liu.

**Writing – review & editing:** Mingjian Kong, Yan Tang, Fei Tong, Xin Lei Zhang, Hua Ni.

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
