## [Decision Letter · Decision Letter 0]

8 Jun 2022

PONE-D-22-10759The analgesic efficacy of pericapsular nerve group block in patients with intertrochanteric femur fracture: a randomized controlled trialPLOS ONE

Dear Dr. Mingjian Kong 

Thank you for submitting your manuscript to PLOS ONE. After careful consideration, we feel that it has merit but does not fully meet PLOS ONE’s publication criteria as it currently stands. Therefore, we invite you to submit a revised version of the manuscript that addresses the points raised during the review process.

I would appreciate if you pay careful attention to the reviewers' comments in your response.==============================

We look forward to receiving your revised manuscript.

Kind regards,

Ehab Farag, MD FRCA FASA

Academic Editor

PLOS ONE

Journal Requirements:

3. We note you have included a table to which you do not refer in the text of your manuscript. Please ensure that you refer to Table 4 in your text; if accepted, production will need this reference to link the reader to the Table.

Reviewers' comments:

Reviewer's Responses to Questions

**Comments to the Author**

1. Is the manuscript technically sound, and do the data support the conclusions?

Reviewer #1: Partly

Reviewer #2: Yes

2. Has the statistical analysis been performed appropriately and rigorously? 

Reviewer #1: No

Reviewer #2: Yes

3. Have the authors made all data underlying the findings in their manuscript fully available?

Reviewer #1: No

Reviewer #2: Yes

4. Is the manuscript presented in an intelligible fashion and written in standard English?

Reviewer #1: Yes

Reviewer #2: Yes

5. Review Comments to the Author

Reviewer #1: This is an interesting study comparing Peripheral nerve blocks to Fascia iliaca compartment block. The manuscript to structured in a easy to read manner.

They a few comments worth addressing.

1. At the end of introduction, can the authors state the primary and secondary objectives.

2. Can the randomisation information include the allocation ratio.

3. Can the authors define their population of analysis for the primary analysis (e.g intent to treat), secondary and safety.

4. Table 1, as this this is an RCT, its recommended not to formally test baseline characteristics, any difference are simply due to chance and also do not serve a useful purpose so p-values should be omitted (see "Comparisons against baseline within randomised groups are often used and can be highly misleading by Bland JM, Altman 2011").

5. Any correction taken into account for multiple testing, i.e testing in each timepoint, increase you chance of finding a significant result?

6. Can the authors state whether a statistical analysis plan was signed off prior to unblinding?

7. Was there an oversight committee to monitor the safety of patients, its not mentioned in the manuscript?

Reviewer #2: Why do you think there was zero incidence of quadriceps motor weakness in the PENG block group compared to prior referenced studies that showed some degree of motor block, although still lower than fascia iliaca? You attributed it related to lower concentration or was the small sample size not adequately powered to detect secondary outcome measurements? Also would have been useful to measure quadriceps muscle strength at timepoints later than 6 hours

How was the method of passive exercise testing decided on? Is there a basis for using a 15 degree passive hip elevation?

How long after block performance were patients induced and surgery started? As t1 is 30 min post block time, does an increased length of time between block performance and starting the timer at the conclusion of surgery influence block duration when measured postop? The length of surgical time is shown but not the length of time between T1 and T2.

Source 13 does not refer to anything relevant to this study - need to edit this

6. PLOS authors have the option to publish the peer review history of their article (what does this mean?). If published, this will include your full peer review and any attached files.

Reviewer #1: No

Reviewer #2: No

---

## [Author Response · Author response to Decision Letter 0]

1 Aug 2022

Editor

Respected Editors：thanks for your review , modifications have been made in the manuscript as requested.

Respected Editors：we have uploaded a separate minimal data.

3. We note you have included a table to which you do not refer in the text of your manuscript. Please ensure that you refer to Table 4 in your text; if accepted, production will need this reference to link the reader to the Table.

Respected Editors：we have modified this and marked it in the manuscript.

Respected Editors：We have added the title of the supporting information file to the end of the manuscript.

5.Please review your reference list to ensure that it is complete and correct. If you have cited papers that have been retracted, please include the rationale for doing so in the manuscript text, or remove these references and replace them with relevant current references. Any changes to the reference list should be mentioned in the rebuttal letter that accompanies your revised manuscript. If you need to cite a retracted article, indicate the article’s retracted status in the References list and also include a citation and full reference for the retraction notice.

Respected Editors：We have checked the reference list and made corrections to ensure it is complete and correct.(all revisions have been marked in the manuscript)

Reviewer 1

1.At the end of introduction, can the authors state the primary and secondary objectives.

It has been supplemented in the introduction section of the manuscript.We will be happy to edit the text further, based on helpful comments from the reviewers. 

2.Can the randomisation information include the allocation ratio.

It has been supplemented in the Randomization and Blinding section of the manuscript.We will be happy to edit the text further, based on helpful comments from the reviewers. 

3.Can the authors define their population of analysis for the primary analysis (e.g intent to treat), secondary and safety.

It has been supplemented in the discussion section of the manuscript.We will be happy to edit the text further, based on helpful comments from the reviewers. 

4.Table 1, as this this is an RCT, its recommended not to formally test baseline characteristics, any difference are simply due to chance and also do not serve a useful purpose so p-values should be omitted (see "Comparisons against baseline within randomised groups are often used and can be highly misleading by Bland JM, Altman 2011").

We thank the reviewer for pointing out thi issue.We studied the document you provided and found that there was a misunderstanding. The baseline written on the form was a clerical error and has been deleted in the text. In our research groups were be compared directly by two-sample methods.

5.Any correction taken into account for multiple testing, i.e testing in each timepoint, increase you chance of finding a significant result?

We thank the reviewer for pointing out thi issue. We have taken this situation into consideration,VAS score belongs to grade data and does not conform to the normal distribution, so we use the median (interquartile spacing) for statistical description, and use Kruskal Wallis H test to get the result P < 0.05. It can be considered that the distribution of scores in each group is not the same, and the difference is statistically significant, and then compare them at different time points to find out the positive results.

6.Can the authors state whether a statistical analysis plan was signed off prior to unblinding?

This study developed a statistical analysis plan before Unblinding. The staff who conducted the data analysis did not understand the processing methods of each group. They only selected the corresponding statistical analysis method according to the type of data collected, and then unblinded after completing the statistical analysis of each group of data.

7. Was there an oversight committee to monitor the safety of patients, its not mentioned in the manuscript?

The Ethics Committee supervised the implementation of this study and protected the rights and interests of patients. Relevant contents have been supplemented in the study design and subjects section of the manuscript.

Reviewer 2

Why do you think there was zero incidence of quadriceps motor weakness in the PENG block group compared to prior referenced studies that showed some degree of motor block, although still lower than fascia iliaca? You attributed it related to lower concentration or was the small sample size not adequately powered to detect secondary outcome measurements? Also would have been useful to measure quadriceps muscle strength at timepoints later than 6 hours.

We thank the reviewer for pointing out thi issue.We consider that this situation may be the reason for the concentration. The reference study nerve block used 0.5% ropivacaine, and this study used 0.375% ropivacaine.

In addition to the time point of 6 hours after operation, we also measured the quadriceps femoris muscle strength at 12 hours and 24 hours after operation, but there was no positive result, so it was not included in the result.

How was the method of passive exercise testing decided on? Is there a basis for using a 15 degree passive hip elevation?

We refer to reference 15 for the specific methods of passive motion testing.We will be happy to edit the text further, based on helpful comments from the reviewers.

How long after block performance were patients induced and surgery started? As t1 is 30 min post block time, does an increased length of time between block performance and starting the timer at the conclusion of surgery influence block duration when measured postop? The length of surgical time is shown but not the length of time between T1 and T2.

We thank the reviewer for pointing out thi issue.We made statistics on T1-T2 of patients in each group, and the results were not statistically significant. Table1 has been supplemented with relevant data.

---

## [Decision Letter · Decision Letter 1]

26 Sep 2022

The analgesic efficacy of pericapsular nerve group block in patients with intertrochanteric femur fracture: a randomized controlled trial

PONE-D-22-10759R1

Dear Dr. Kong

We’re pleased to inform you that your manuscript has been judged scientifically suitable for publication and will be formally accepted for publication once it meets all outstanding technical requirements.

Kind regards,

Rizaldy Taslim Pinzon

Academic Editor

PLOS ONE

Additional Editor Comments (optional):

Thank you for addressing comments from the previous submission. The included tables and figures bring added clarity to the data and results.

Reviewers' comments:

Reviewer's Responses to Questions

**Comments to the Author**

1. If the authors have adequately addressed your comments raised in a previous round of review and you feel that this manuscript is now acceptable for publication, you may indicate that here to bypass the “Comments to the Author” section, enter your conflict of interest statement in the “Confidential to Editor” section, and submit your "Accept" recommendation.

Reviewer #2: All comments have been addressed

2. Is the manuscript technically sound, and do the data support the conclusions?

Reviewer #2: Yes

3. Has the statistical analysis been performed appropriately and rigorously? 

Reviewer #2: Yes

4. Have the authors made all data underlying the findings in their manuscript fully available?

Reviewer #2: Yes

5. Is the manuscript presented in an intelligible fashion and written in standard English?

Reviewer #2: Yes

6. Review Comments to the Author

Reviewer #2: Thank you for addressing comments from the previous submission. The included tables and figures bring added clarity to the data and results.

One comment would be for the addition to the beginning of the discussion section: "We also

analyzed the safety of anesthesia related complications after nerve block surgery" - this sentence is unclear, the word surgery probably should not be included after nerve block.

7. PLOS authors have the option to publish the peer review history of their article (what does this mean?). If published, this will include your full peer review and any attached files.

Reviewer #2: No

---

## [Editor Report · Acceptance letter]

2 Oct 2022

PONE-D-22-10759R1 

*The analgesic efficacy of pericapsular nerve group block in patients with intertrochanteric femur fracture: a randomized controlled trial*

Dear Dr. Kong:

I'm pleased to inform you that your manuscript has been deemed suitable for publication in PLOS ONE. Congratulations! Your manuscript is now with our production department. 

Kind regards, 

on behalf of

Dr. Rizaldy Taslim Pinzon 

Academic Editor

PLOS ONE